# Apolipoprotein E Gene in α-Synucleinopathies: A Narrative Review

**DOI:** 10.3390/ijms25031795

**Published:** 2024-02-01

**Authors:** Ioannis Liampas, Panagiota Kyriakoulopoulou, Vasileios Siokas, Eirini Tsiamaki, Polyxeni Stamati, Zinovia Kefalopoulou, Elisabeth Chroni, Efthimios Dardiotis

**Affiliations:** 1Department of Neurology, University Hospital of Larissa, School of Medicine, University of Thessaly, 41100 Larissa, Greece; vsiokas@med.uth.gr (V.S.); tzeni_0@yahoo.gr (P.S.); edar@med.uth.gr (E.D.); 2Department of Neurology, University Hospital of Patras, School of Medicine, University of Patras, 26504 Rio Patras, Greece; panagiotakyriak@yahoo.com (P.K.); eirinitsiamaki@gmail.com (E.T.); zkefalopoulou@gmail.com (Z.K.); echroni@upatras.gr (E.C.)

**Keywords:** Parkinson’s disease, Parkinson’s disease dementia, dementia with Lewy bodies, Lewy body dementia, multiple system atrophy

## Abstract

In this narrative review, we delved into the intricate interplay between *Apolipoprotein E (APOE)* alleles (typically associated with Alzheimer’s disease—AD) and alpha-synucleinopathies (aS-pathies), involving Parkinson’s disease (PD), Parkinson’s disease dementia (PDD), dementia with Lewy bodies (DLB), and multiple-system atrophy (MSA). First, in-vitro, animal, and human-based data on the exacerbating effect of *APOE4* on LB pathology were summarized. We found robust evidence that *APOE4* carriage constitutes a risk factor for PDD—APOE2, and APOE3 may not alter the risk of developing PDD. We confirmed that *APOE4* copies confer an increased hazard towards DLB, as well. Again *APOE2* and *APOE3* appear unrelated to the risk of conversion. Of note, in individuals with DLB *APOE4*, carriage appears to be intermediately prevalent between AD and PDD-PD (AD > DLB > PDD > PD). Less consistency existed when it came to PD; *APOE*-PD associations tended to be markedly modified by ethnicity. Finally, we failed to establish an association between the *APOE* gene and MSA. Phenotypic associations (age of disease onset, survival, cognitive–neuropsychiatric- motor-, and sleep-related manifestations) between *APOE* alleles, and each of the aforementioned conditions were also outlined. Finally, a synopsis of literature gaps was provided followed by suggestions for future research.

## 1. Introduction

In the intricate field of neurodegeneration, the pathogenesis of alpha-synucleinopathies (aS-pathies) remains enigmatic. Although a delicate interplay between environmental factors and genetics is theorized to be accountable, the etiology of aS-pathies is far from unveiled. Pathologically, aS-pathies are characterized by aggregations of a protein known as alpha-synuclein (aS) within neurons and/or supporting brain cells [1]. These neuronal and/or glial inclusions contribute to neuronal damage and depending on their territorial distribution they may manifest with various phenotypes including Parkinson’s disease (PD), Parkinson’s disease dementia (PDD), dementia with Lewy bodies (DLB), and multiple-system atrophy (MSA) [2]. The *apolipoprotein E* (*APOE*) gene is involved in the construction of APOE, a multifunctional protein integral to lipid metabolism and transport [3]. *APOE* genotypes have well-established risk-modifying properties in Alzheimer’s disease (AD) and appear to be implicated in the pathology of aS-pathies, as well [4,5,6,7]. The current narrative review aims to plunge into the puzzling interactions between the different *APOE* alleles and aS-pathies, seeking to provide novel insights into the molecular foundations of these neurodegenerative entities. Published evidence on the role of *APOE* genotypes in terms of aS-pathies’ prevalence, incidence, or other important disease parameters (e.g., phenotypic variations, disease severity, mortality rates, and so on) will be summarized, while literature gaps and areas of contradiction will be untangled and discussed.

### 1.1. a-Synucleopathies

A neurodegenerative disorder characterized by the accumulation of the aS protein aggregates within nerve cells and/or supporting brain cells is defined aS-pathy. Identified in 1988 by Maroteaux and colleagues, aS is coded by the *Synuclein Alpha* (*SNCA*) gene on the long arm of chromosome 4 (4q21) [8,9]. aS is predominantly abundant in presynaptic nerve terminals, presumably assuming a pivotal role in synaptic functions, synaptic plasticity, and neurotransmission [10]. 

Lewy bodies (LBs) and Lewy neurites (LNs) compose the defining pathological markers of aS-pathies. In 1997, it was observed that aS is present in Lewy bodies (LBs), which additionally contain other proteins, such as ubiquitin, neurofilament protein, and alpha B crystallin [11]. Although the complete process of LB formation remains a mystery, it is believed that interactions between aS monomers and lipid membranes as well as the compromization of stable aS tetramers constitute critical steps towards oligomerization and in turn aggregation of aS [12,13]. Lewy neurites (LNs)—the second hallmark of aS-pathies—constitute dystrophic neuritic processes in degenerating neurons featuring the same immunohistochemical profile as LBs [14]. 

aS-pathies are broadly classified into LB disease and MSA; the three core phenotypes of LB disease are PD, PDD, and DLB; the two cardinal clinicopathologic subtypes of MSA are the parkinsonian type (MSA-P) and the cerebellar type (MSA-C) [1]. In LB disease, aS aggregation and LB formation primarily affect neurons, whereas in MSA glial cytoplasmic inclusions (GCIs) are principally configured [15,16]. These entities share similar signs and symptoms, and their clinical distinction remains quite challenging even after the implementation of more elaborate and sophisticated investigations [17]. 

### 1.2. Parkinson’s Disease

The first description of PD is dated back to 1817, by dr. James Parkinson [18]. However, it was not until 1912 that dr. Friedrich H. Lewy found intracytoplasmic inclusions in the brains of deceased patients with PD; these neuronal inclusions were shortly later named after him [1,18]. Nearly two centuries after its initial description in 1988, the United Kingdom RD Society Brain Bank (UKPDSBB) introduced the first formal criteria for the clinical diagnosis of PD [19]. 

PD is a progressive neurodegenerative disorder of the central nervous system (CNS) marked by cardinal movement manifestations involving resting tremor, rigidity, bradykinesia, and postural instability [20]. Autonomic dysfunction, anosmia, sleep, cognitive, and neuropsychiatric symptoms may occur. PD is associated with degeneration of dopamine-producing neurons in the pars compacta of the substantia nigra [21]. Cytoplasmic inclusions of aS forming LBs and LNs tend to accumulate within affected neurons [2,11,21]. Following AD, it is the most common neurodegenerative disorder as well as the most prevalent entity among aS-pathies [2].

### 1.3. Lewy Body Dementia

In 1962, dr. John Woodard documented a series of cases with prominent neuropsychiatric manifestations and predominant LB pathology in brain autopsies [22]. A fraction of these patients exhibited Parkinsonian features, as well. Additional reports from Japan were published over the next decades: cases with LB pathology of variable distributions, associated with heterogeneous phenotypes including motor, neurocognitive, or neuropsychiatric symptoms were detailly described [15]. In 1996, the first consensus guidelines for the diagnosis of DLB were published [23]. Today, the broader term LBD is used to encompass DLB and PDD, two major neurocognitive entities that present substantial clinicopathologic and neurochemical overlap. Their distinction is rather arbitrary, with the ‘one-year rule’ groundlessly distinguishing between the two [24]. If dementia coincides with or emerges within one year after the occurrence of Parkinsonism, DLB is diagnosed; if Parkinsonism precedes dementia onset by more than one year, the diagnosis of PDD is established [25].

LBD primarily manifests with cognitive, neuropsychiatric, motor (parkinsonism), and sleep disturbances [26]. The cardinal cognitive manifestations involve executive, visuospatial, and attention deficits, manifesting with a fluctuating course. Visual hallucinations are the most characteristic among neuropsychiatric symptoms, with delusions, hallucinations in other modalities, apathy, and affective disorders ensuing. REM sleep behavior disorder (RBD) has been integrated into the core clinical diagnostic criteria while hypersomnia constitutes a supportive feature. Hyposmia, autonomic dysfunction, and hypersensitivity to neuroleptics, among others, are supportive clinical manifestations.

LBD is a quite prevalent in degenerative major neurocognitive disorders, second only to AD [27]. The pathological hallmark of LBD is the presence of LBs and LNs in the brainstem, limbic system, and cerebral cortex [28,29]. Given the absence of specific therapies, understanding the pathobiology of LBD is crucial for the development of new treatments. 

### 1.4. Multiple System Atrophy

The first report of two cases with MSA (formerly known as Shy Drager syndrome) is attributed to dr. Milton Shy and dr. Glen Drager back in 1960. Almost four decades later, in 1998, the first consensus diagnostic criteria were published, separating the Parkinsonian from the cerebellar MSA type [30,31]. 

MSA is a rare neurodegenerative disorder that presents with autonomic dysfunction and either predominant Parkinsonian features (poorly responsive to dopamine replacement) or prominent cerebellar syndrome [32]. Symptoms such as RBD, dysphagia, speech impairment, respiratory stridor, or olfactory dysfunction may also coexist. Pathologically, GCIs of aS are accumulated in the olivopontocerebellar and/or striatonigral system [24,33]. Cognitive changes tend to be of secondary importance in MSA.

## 2. The Multifaceted Role of Apolipoprotein E in the Brain

*APOE* has three major polymorphic alleles in humans (*APOE2*, *APOE3*, and *APOE4*). Each allele is related to important structural and functional alterations in the proteinic molecule of APOE [34]. APOE is an extracellular protein synthesized by astrocytes (primarily responsible for its production) and activated microglia in the brain. It plays a pivotal role in brain homeostasis via various pathways, including lipid transport, glucose metabolism, synaptic integrity, and plasticity, as well as membrane trafficking [35,36]. The relationship between APOE isoforms and AD risk is well-established [4,5]. Each APOE isoform appears to be differentially associated with amyloid β (Aβ)-related and Aβ-independent pathways involved in the course of AD (e.g., neuroinflammation, vascular function, blood–brain barrier function, and so on), ultimately altering the net risk of incident AD [34]. Previous research has specifically shown the crucial role of APOE in the metabolism of Aβ [37,38,39,40]. Isoform-dependent binding to Aβ regulates its production and clearance. APOE4 enhances Aβ production and hinders its phagocytic clearance leading to Aβ deposition. APOE2, on the other hand, decelerates this process. At the same time, the role of APOE–lipid interactions appears to be of pivotal importance in αS aggregation [41]. Neuronal APOE has been reported to attenuate both neuronal αS uptake and release, with APOE deficiency decreasing the expression of APOE receptors responsible for αS uptake and enhancing chaperone-mediated autophagy [42]. APOE deficiency ultimately results in the accumulation of insoluble αS and phosphorylated αS in the brain, as well as altered membrane lipid profiles [36]. The modification of membrane composition appears to influence αS binding which might lead to altered β-sheet formation and, in turn, further fibrillization [36]. 

### 2.1. Clinical Relationship of APOE with Alzheimer’s Disease 

AD stands as the predominant cause of dementia globally with its prevalence surging on the grounds of the increasingly prolonged life expectancy. Amid the array of identified risk factors, *APOE* genotypes emerge as the most important genetic determinants of late-onset AD [4]. *APOE3*—the most common genetic variant—is considered neutral in terms of incident AD risk [43]. *APOE4* variants confer an elevated hazard towards late-onset sporadic AD in a dose-dependent manner (risk size increases relative to the number of *APOE4* alleles) [4,5,43]. At the same time, *APOE4* is linked to a younger age of (late-onset) AD onset, with this effect again varying as a function of the number of *APOE4* copies [5]. On the other hand, *APOE2* has a protective effect against AD. Individuals carrying one or two copies of the *APOE2* have a reduced dose-dependent risk of developing AD; those who do convert to AD tend to do so at an older age [4,5,43,44]. The above-mentioned association between *APOE* alleles and susceptibility to AD (APOE*4* > *3* > *2*) is mediated via multiple pathways: increased Aβ deposition and tau aggregation, induction of neuroinflammation through the production of proinflammatory cytokines and microglia stimulation, increased intracellular lipid accumulation, and disruption of effective myelin formation (among others) [45]. In addition to these common alleles, several rare *APOE* variants, such as *apoE3-R136S* (known as *apoE3-Christchurch* or *apoE3-Ch*), *apoE3-V236E* (referred to as *apoE3-Jacksonville* or *apoE3-Jac*), and *apoE4-R251G*, have been identified. These rare variants are believed to offer some level of protection against the pathological processes associated with AD [4]. 

### 2.2. Associations between APOE and a-Synuclein Pathology

Apart from the two primary pathological features—Aβ and tau depositions—AD brains often exhibit additional pathological alterations [35]. Large autopsy series of patients with a clinically established diagnosis of AD have revealed that only a fraction of AD cases (between 35 and 50%) show pure AD pathology: most cases exhibit mixed neuropathological alterations with predominant vascular (~25%), LB (~13%), or other (e.g., TDP43) specific pathologies [46,47,48,49]. These findings have given rise to theories of genetic overlap between AD and LB pathology and have fueled relevant research [50]. In this context, in the last few years, a number of studies have tried to shed light on the relationship between the *APOE* gene and LB pathology, in association with or independently of the presence of AD pathology.

The first reports that provided preliminary pathologoanatomic evidence consistent with the hypothesis that the *APOE* gene is related to LB pathology are dated back to 1995 [51]. Later, interactions between intracellular aS and APOE (protein) dependent pathways were suggested to mediate the stimulation of shared neurodegenerative mechanisms in PD and AD [52]. In the course of time, additional evidence has accumulated to confirm the role of *APOE* gene in LB pathology and emplace *APOE* in its rightful spot among genetic factors with proven importance in the field of aS-pathies: (1) Emamzadeh and colleagues revealed that *APOE4* was linked to aS aggregation, using in vitro models; [53]; (2) Zhao and colleagues found that *APOE4* exacerbated αS pathology (as well as astrogliosis, neuronal, and synaptic loss) independently of Aβ deposition, using both animal models and postmortem human brains [36,54]; (3) Davis and colleagues replicated these findings in animal models while added some evidence on a potential protective role of *APOE2* against aS aggregation [41]; (4) Mann and colleagues as well as Dickson and colleagues showed that *APOE4* carriage leads to greater severity of LB pathology in autopsy confirmed cases of DLB [55,56]; (5) Gearing and colleagues revealed that a dose-dependent association exists between *APOE4* (as a function of the number of copies) and PD-related pathological changes in neuropathologically confirmed AD cases [57]; (6) Wakabayasi and colleagues reported that both LB and AD pathology are increased in PD carriers of *APOE4* [28]; (7) Jin and colleagues found that *APOE4* increases LB pathology in brains of autopsy confirmed AD patients [35]; (8) Robinson and colleagues exhibited that *APOE4* is a risk factor for co-pathologies independent of neurodegenerative disease, with Aβ and aS being most prevalent [58].

On the other hand, researchers have occasionally found that *APOE4* is linked to concomitant AD pathology among cases with LB pathology, but patients with pure LB neuropathologic alterations have similar *APOE4* carriage frequencies to those without LB/AD pathology [59,60,61,62]. Additionally, *APOE4* alleles have been reported to lead to earlier onset of neuropathologically confirmed mixed AD/LB—but not pure LBD—dementia in a dose-dependent manner [63]. Of note, these findings do not preclude an association between *APOE4* and LB histopathology since the strong well-established relationship between *APOE4* and AD is probably overwhelming that between *APOE4* and LB pathology, introducing predominantly AD-related co-pathological changes in the vast majority of cases. Hence, reports of an association between *APOE4* and LBD in the absence of (at least mild) AD co-pathology are rare [64,65].

With respect to PD in particular, there are more studies suggesting that *APOE4* is distributed similarly to non-PD controls; also, *APOE4* seems to increase only concomitant AD-related neuropathologic alterations or cortical—but not nigral—LB pathology [66,67,68]. Therefore, the scarcity of supporting evidence corroborates a lack of an association between PD and *APOE4*. The induction of AD or cortical LB neuropathology may, however, account for some phenotypic variation irrespective of the major underlying neurodegenerative cause (e.g., PD or MSA) [69,70]. Of course, apart from AD-LB neuropathology, latent associations or interactions between *APOE* alleles and alternative pathologies that mediate phenotypic variations cannot be excluded. As a paradigm serves the relationship of *APOE* with white matter hyperintensities (WMH). For instance, significant interactions between WMH burden and *APOE4* carriage were found to mediate cognitive performance in older adults with AD or DLB: WMH volume was associated with poorer cognitive performance (attention, executive function, memory, and language) only in *APOE4* carriers [71]. 

### 2.3. Clinical Links between APOE-PD and PDD

*APOE* has been a subject of significant interest in the field of PD research (Table 1). Understanding its role in PD has been a complex endeavor, with studies presenting variable and sometimes conflicting findings [72]. Hence, the exact impact of *APOE* on PD remains a topic of ongoing investigation and debate within the scientific community. 

In 2009, Williams-Gray and colleagues updated the original meta-analyses of Huang and colleagues on the associations between *APOE* alleles and PD or PDD [73,74,75]. Synthesizing the results of 32 case–control studies, the authors found that the presence of at least one *APOE2* allele contributed modestly to PD susceptibility (OR = 1.16). On the other hand, using data from 17 case–control studies, *APOE4* carriage was reported to confer an elevated hazard towards PDD by a more prominent association (OR = 1.74). About a decade later, Li and colleagues pooled data from 47 case–control studies and replicated the modest association between *APOE2* carriage and PD (OR = 1.23) [76]. At the same time, subgroup analyses revealed a new association between *APOE4* and PD, limited only among individuals of Asian ancestry (OR = 1.43). The authors looked into genotypic associations as well: *APOE2/4* genotype was found to confer a substantial risk towards PD in Asians (OR = 4.43) and *APOE3/4* was reported to constitute a moderate risk factor for PD among Latin-American populations (OR = 1.44) and exert a protective effect against PD among Caucasians (OR = 0.86). Shortly after, the ethnic association between *APOE4* and PD was reproduced in the meta-analysis of Sun and colleagues (39 case–control studies) [77]. Investigators found that *APOE2* and *APOE4* are not related to the risk of PD—the only exception being Asian populations where *APOE4* was found to modestly increase the risk of PD (OR = 1.22). Moreover, *APOE3* showed a mild protective effect against PD [OR = 0.90]. The risk of PDD on the other hand was found elevated in those with *APOE4* [(OR = 1.46), an association that was accentuated in Asian populations (OR = 1.88)] and moderated among individuals with *APOE3* (OR = 0.72). Finally, in 2018, in their meta-analysis of case–control studies (17 in total), Pang and colleagues confirmed the relationship between *APOE4* and PDD (OR = 1.72) and the lack of a relationship between *APOE2*-*3* and PDD [78]. Overall, all meta-analyses agree that *APOE4* confers a risk towards PDD. Less consistency exists when it comes to PD. Incongruent evidence leans towards modest protective properties for *APOE3* and differential *APOE2* and *APOE4* (or genotypic) associations by ethnicity. 

Apart from susceptibility to PD and PDD, several researchers have focused on the potential link between *APOE4* and the age of PD onset. Published evidence has occasionally suggested that the presence of at least one *APOE4* copy is related to earlier PD onset while the presence of *APOE3* and/or *APOE2* alleles may delay its onset [79,80,81,82,83]. At the same time, the vast majority of published reports failed to reproduce these associations precluding any relationship between *APOE* and age of PD onset [84,85,86,87,88,89].

Another aspect that has accumulated considerable interest is the potential phenotypic associations of *APOE4* and PD. Several researchers have found that *APOE4* carriage is related to steeper cognitive decline and especially memory and executive function decline [90,91,92,93,94,95,96,97]. Similarly, *APOE4* alleles have been reported to contribute to more severe motor semiology and more abrupt motor progression (e.g., more common gait freezing, higher UPDRS total scores, and more precipitous motor decline) [98,99,100,101]. Finally, *APOE4* carriers have a stronger affinity towards neuropsychiatric manifestations (especially psychotic symptoms) [98,101,102,103]. Again, published evidence is not uniformly concurring with respect to neuropsychiatric and motor associations; however, the consistency and reproducibility of cognition-related findings probably reflect a true relationship between *APOE4* and cognitive impairment—decline in individuals with PD [18,94,104,105,106]. By extension, these findings come in accordance with the robust relationship between *APOE4* and PDD.

### 2.4. Clinical Relationship between APOE-DLB

In 2020, the meta-analysis by Sanghvi and colleagues (synthesizing data from 75 articles in total) confirmed the association between *APOE4* carriage and DLB (OR = 2.70) while replicating the weaker, already known association between *APOE4* and PDD (OR = 1.60) [107] (Table 1). Of note, *APOE4* copies appear to be intermediately prevalent between AD and PDD-PD (AD > DLB > PDD > PD), although a minority of papers report a prevalence even higher than in AD [64,108,109,110,111,112,113].

Less evidence exists with respect to the age of DLB onset, with researchers occasionally reporting an association between *APOE4* and earlier age of DLB onset (similar to AD) and only one report showing that *APOE2* may delay conversion to DLB [108,114,115]. Furthermore, *APOE4* copies have been related to a dose-dependent decrease in survival among individuals with DLB (similar to those with AD) [50,116,117,118,119]. 

Regarding phenotypic associations, *APOE4* expression in aS animal (mice) models have been related to impaired cognitive and behavioral performances [54]. Moreover, the presence of at least one *APOE4* copy among DLB patients has been linked to cognitive, neuropsychiatric, and autonomic manifestations involving steeper cognitive decline, memory and executive dysfunction, delusions, apathy, depression, and hyperhidrosis but not any motor symptoms [62,113,120]. 

### 2.5. Clinical Associations between APOE-MSA

Cairns and colleagues were the first to investigate the relationship between *APOE* gene and MSA and reported that *APOE4* was equally prevalent between individuals with MSA and healthy controls [121]. Shortly after, the lack of an association between *APOE* alleles and MSA was replicated in other case–control studies) [118,122,123] while evidence was also added on the absence of an association with age of MSA onset [124] (Table 1). Of note, APOE4 has been additionally found unrelated to the risk of idiopathic RBD conversion to aS-pathies (PB DLB or MSA) [125]. Lately, although an association with MSA again failed to be established, researchers have reported signs of an effect of *APOE4* on reduced aS uptake from oligodendroglia among adults with MSA [126,127]. Irrespective of the lack of a link between *APOE4* and MSA, research is generally scarce with respect to the potential effect of *APOE4* on phenotypic MSA variations (e.g., a recent study found that *APOE4* carriage may be associated with depression in MSA carriers) [126].

**Table 1 ijms-25-01795-t001:** Summary of clinical associations between APOE alleles and α-Synucleinopathies.

PD	*APOE* by ethnic interactions may alter PD risk: *APOE4* may confer a risk towards PD in Asian populations (OR = 1.22—Sun et al., 2019 [77]; OR = 1.43—Li et al., 2018 [76]), *APOE2/4* genotype may increase the risk more prominently among Asians (OR = 4.43—Li et al., 2018 [76]), *APOE3/4* may constitute a risk factor for PD in Latin-American populations (OR 1.44—Li et al., 2018 [76]) and exert a protective effect against PD among Caucasians (OR = 0.86—Li et al., 2018 [76]). Age of PD onset is probably unrelated to *APOE* alleles. *APOE4* carriage is related to steeper cognitive decline. *APOE4* copies may elevate the risk of neuropsychiatric manifestations. The association between *APOE* alleles and motor progression requires further research.
PDD	*APOE4* carriage confers an elevated hazard towards PDD (OR = 1.60—Sanghvi et al., 2020 [107]; OR = 1.72—Pang et al., 2018 [78]; OR = 1.74—Williams-Gray et al., 2009 [75]). Ethnic interactions may play a role: *APOE4* may confer an elevated risk towards PD in Asian populations (OR = 1.88 Sun et al., 2019 [77])
DLB	*APOE4* carriage confers an elevated hazard towards DLB (OR = 2.70—Sanghvi et al., 2020 [107]). *APOE4* copies may decrease survival among individuals with DLB. Age of DLB onset and phenotypic associations require further research.
MSA	Evidence suggests against an association between *APOE* alleles and MSA. Phenotypic associations require further research.

PD: Parkinson’s disease; PDD: PD dementia; DLB: dementia with Lewy bodies; MSA: multiple system atrophy; *APOE: apolipoprotein E*; OR: odds ratio.

## 3. Literature Gaps and Future Perspectives

Looking ahead, further research is warranted to decipher the interplay between the *APOE* gene and aS-pathies. Future studies should venture deeper into the intricate molecular mechanisms through which *APOE4* drives neurodegeneration, expedites aS aggregation and configures its territorial distribution (cortical vs. nigral), modulates neuroinflammation, and affects amyloid and tau deposition in individuals with LBD. Of note, shared genetic loci between AD and LBD probably reflect the existence of common neurodegenerative pathways. Therefore, relevant research promises not only enhanced comprehension of the molecular and pathophysiological foundations of LBD but also a broader illumination of the mechanisms underlying neurodegeneration. 

Further studies are also warranted to shed light on the relationship between *APOE4* and LBD’s phenotypic variations. Focus should be placed on the potential association between *APOE4* and cognitive decline to elucidate which specific domains are predominantly affected and which of these associations are driven by aS or AD-related co-pathological changes. Moreover, apart from psychotic symptoms, published studies have not investigated phenotypic associations of *APOE4* with neuropsychiatric manifestations. Again, considering that a pathological AD component is to be expected, researchers ought to include cases with available brain autopsies. In addition, contradictory evidence exists with respect to the *APOE* gene—motor associations; therefore, upcoming studies should provide more definitive conclusions. In the same context, various other phenotypic features of LBD (e.g., RBD, autonomic dysfunction, and neuroleptic sensitivity) remain almost utterly unexplored in terms of association with *APOE4* and future articles shall delve into these potential associations, as well.

## 4. Conclusions

This narrative literature review delved into the multifaceted role of the *APOE* gene in aS-pathies. We found robust evidence that *APOE4* carriage constitutes a risk factor for PDD—*APOE2* and *APOE3* may not alter the risk of progression. We confirmed that *APOE4* copies confer an increased hazard towards DLB, as well. Again, *APOE2* and *APOE3* appear unrelated to the risk of conversion. Of note, in individuals with DLB, APOE4 carriage appears to be intermediately prevalent between AD and PDD-PD (AD > DLB > PDD > PD). Less consistency existed when it came to PD while the *APOE* gene–PD associations tended to be markedly modified by ethnicity. Finally, we failed to establish an association between the *APOE* gene and MSA. In terms of phenotypic associations, *APOE4* carriers exhibit more precipitous cognitive decline and a tendency towards psychotic manifestations (hallucinations or delusions) irrespective of the exact underlying neurodegenerative entity (PD, PDD, or DLB). Individuals with DLB and *APOE4* also had elevated mortality rates. Motor symptoms and signs, on the other hand, appear to be unrelated to the *APOE* gene. The relationship of *APOE* alleles with the remaining features of aS-pathies remains enigmatic (other neuropsychiatric manifestations, autonomic dysfunction, RBD, neuroleptic sensitivity, and so on).

aS-pathies, with their complex clinical manifestations and elusive nature, persist as a formidable diagnostic and therapeutic challenge. As the armamentarium and availability of more elaborate diagnostic biomarkers increase, the pre-mortem diagnosis of these conditions has become more and more accurate. On the other hand, the treatment of these conditions is limited to symptom management without any available agents to intercept their progress. Therefore, a better understanding of aS-pathies’ pathogenesis is of crucial importance so as to identify new treatment targets (specific molecules, pathophysiological pathways, homeostatic mechanisms, and so on). Ongoing investigations on the role of *APOE* in aS-pathies will deepen our understanding of these complex interactions. It remains to be seen, if these findings will transition from mere scientific discoveries to actionable therapies.

## Data Availability

Not applicable.

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
