# Peer review of "Apolipoprotein E Gene in α-Synucleinopathies: A Narrative Review"

_ijms, 2024, doi:10.3390/ijms25031795_

Round 1

Reviewer 1 Report

Comments and Suggestions for Authors

In this manuscript, the authors reviewed the main findings on the impact of ApoE alleles on α-synucleinopathies, including the association between ApoE4 and LB pathology, as well as the increased risk of PDD in ApoE4 carriers. Also, it highlights the potential therapeutic implications of these findings, including the need for personalized treatment approaches based on the ApoE genotype. Furthermore, the review also identified future research directions, such as investigating the role of ApoE in the pathogenesis of α-synucleinopathies and exploring potential interventions to mitigate the impact of ApoE4 on LB pathology. Overall, this review provides valuable insights into the complex interplay between ApoE and α-synucleinopathies, shedding light on potential avenues for future research and treatment. I have only some minor points for the references of the authors.

1.     Tab. 1: This table is organized in a way that is difficult for readers to read and provides limited content. I would suggest listing the odds ratios of the four diseases PD/PDD/DLB/MSA related to ApoE, and labeling them with the corresponding literature. This makes it easier for readers to quickly understand the current general research status.

2.     This manuscript clearly explained the correlation between ApoE and the four diseases of PD/PDD/DLB/MSA. But if they can add another part to explain a little bit how ApoE2/3/4 affects the progression of the disease at the level of molecular mechanisms or cell biology, I think its scientific value will increase.

3.     The English writing is already quite fluent, but the rhetoric in some places could be better.

Comments on the Quality of English Language

Only needs very slight rhetorical polish.

Author Response

Reviewer #1

Comments and suggestions for authors:

In this manuscript, the authors reviewed the main findings on the impact of ApoE alleles on α-synucleinopathies, including the association between ApoE4 and LB pathology, as well as the increased risk of PDD in ApoE4 carriers. Also, it highlights the potential therapeutic implications of these findings, including the need for personalized treatment approaches based on the ApoE genotype. Furthermore, the review also identified future research directions, such as investigating the role of ApoE in the pathogenesis of α-synucleinopathies and exploring potential interventions to mitigate the impact of ApoE4 on LB pathology. Overall, this review provides valuable insights into the complex interplay between ApoE and α-synucleinopathies, shedding light on potential avenues for future research and treatment. I have only some minor points for the references of the authors.

Response: Thank you for your valuable comments and suggestions. A point-by-point detailed response is provided below.

Tab. 1: This table is organized in a way that is difficult for readers to read and provides limited content. I would suggest listing the odds ratios of the four diseases PD/PDD/DLB/MSA related to ApoE, and labeling them with the corresponding literature. This makes it easier for readers to quickly understand the current general research status.

Response: Thank you for your comment. Table 1 was modified as suggested.

This manuscript clearly explained the correlation between ApoE and the four diseases of PD/PDD/DLB/MSA. But if they can add another part to explain a little bit how ApoE2/3/4 affects the progression of the disease at the level of molecular mechanisms or cell biology, I think its scientific value will increase.

Response: Thank you for your comment. We added a relevant paragraph in the section ‘‘The multifaceted role of apolipoprotein E in the brain’’: Previous research has specifically shown the crucial role of APOE in the metabolism of Aβ (10.1016/j.biopha.2023.116071, 10.1016/j.bbrc.2023.10.038, 10.1016/j.neulet.2023.137532,https://doi.org/10.1007/s00018-023-05026-w). Isoform dependent binding to Aβ regulates its production and clearance. APOE4 enhances Aβ production and hinders its phagocytic clearance leading to Aβ deposition. APOE2, on the other hand, decelerates this process. At the same time, the role of APOE-lipid interactions appears to be of pivotal importance in αS aggregation, as well (https://doi.org/10.1126/scitranslmed.aay3069). Neuronal APOE has been reported to attenuate both neuronal αS uptake and release, with APOE deficiency decreasing expression of APOE receptors responsible for αS uptake and enhancing chaperone-mediated autophagy (https://doi.org/10.3390/ijms23158311). APOE deficiency ultimately results in accumulation of insoluble αS and phosphorylated αS in the brain, as well as altered lipid profiles (10.1007/s00401-021-02361-9). The modification of membrane composition appears to influence αS binding which might lead to altered β-sheet formation and in turn further fibrillization (10.1007/s00401-021-02361-9). 

The English writing is already quite fluent, but the rhetoric in some places could be better.

Response: Thank you for kindly noticing. We edited the manuscript to improve its reading.

Reviewer 2 Report

Comments and Suggestions for Authors

In their paper entitled “Apolipoprotein E Gene in α-Synucleinopathies: a Narrative Review”, the Authors discuss a series of data related to the interplay between Apolipoprotein E (APOE) alleles, normally considered in association with Alzheimer's disease (AD), and the alpha-synucleinopathies, observed in Parkinson's disease (PD), Parkinson's disease dementia (PDD), dementia with Lewy bodies (DLB), and multiple-system atrophy (MSA). 

The paper is of interest and suitable for Int. J. Mol. Sci. Moreover, about 30% of the reported references concerns papers published in the last 5 years (2019-2023). 

However, even if there is not yet any clear idea on the role played by APOE in protein aggregation, I think,  that, in order to improve the interest of the paper, a few mechanisms might be hypothesized by the Authors, just to also give a molecular meaning to the reported information. In particular, lipids, and thus APOE proteins, might have a role in protein aggregation; see, for example the following very recent papers.

1.Wang X. et al. The function of sphingolipids in different pathogenesis of Alzheimer's disease: A comprehensive review. Biomed. Pharmacother. 2024, 171, 116071. doi: 10.1016/j.biopha.2023.116071.

2. Lewkowicz E, et al. Molecular modeling of apoE in complexes with Alzheimer's amyloid-β fibrils from human brain suggests a structural basis for apolipoprotein co-deposition with amyloids. Cell Mol Life Sci. 2023 Nov 27;80(12):376. doi: 10.1007/s00018-023-05026-w.

3. Gholami A. Alzheimer's disease: The role of proteins in formation, mechanisms, and new therapeutic approaches. Neurosci Lett 2023 Nov 20:817:137532. doi: 10.1016/j.neulet.2023.137532.

4.Takebayashi Y, et al. Apolipoprotein E genotype-dependent accumulation of amyloid β in APP-knock-in mouse model of Alzheimer's disease. Biochem Biophys Res Commun 2023 Nov 26:683:149106. doi: 10.1016/j.bbrc.2023.10.038.

Author Response

Reviewer #2

Comments and suggestions for authors:

In their paper entitled “Apolipoprotein E Gene in α-Synucleinopathies: a Narrative Review”, the Authors discuss a series of data related to the interplay between Apolipoprotein E (APOE) alleles, normally considered in association with Alzheimer's disease (AD), and the alpha-synucleinopathies, observed in Parkinson's disease (PD), Parkinson's disease dementia (PDD), dementia with Lewy bodies (DLB), and multiple-system atrophy (MSA). The paper is of interest and suitable for Int. J. Mol. Sci. Moreover, about 30% of the reported references concerns papers published in the last 5 years (2019-2023).                  

Response: Thank you for your valuable comments and suggestions. A point-by-point detailed response is provided below.

However, even if there is not yet any clear idea on the role played by APOE in protein aggregation, I think,  that, in order to improve the interest of the paper, a few mechanisms might be hypothesized by the Authors, just to also give a molecular meaning to the reported information. In particular, lipids, and thus APOE proteins, might have a role in protein aggregation; see, for example the following very recent papers.

1.Wang X. et al. The function of sphingolipids in different pathogenesis of Alzheimer's disease: A comprehensive review. Biomed. Pharmacother. 2024, 171, 116071. doi: 10.1016/j.biopha.2023.116071.

2.Lewkowicz E, et al. Molecular modeling of apoE in complexes with Alzheimer's amyloid-β fibrils from human brain suggests a structural basis for apolipoprotein co-deposition with amyloids. Cell Mol Life Sci. 2023 Nov 27;80(12):376. doi: 10.1007/s00018-023-05026-w.

3.Gholami A. Alzheimer's disease: The role of proteins in formation, mechanisms, and new therapeutic approaches. Neurosci Lett 2023 Nov 20:817:137532. doi: 10.1016/j.neulet.2023.137532.

4.Takebayashi Y, et al. Apolipoprotein E genotype-dependent accumulation of amyloid β in APP-knock-in mouse model of Alzheimer's disease. Biochem Biophys Res Commun 2023 Nov 26:683:149106. doi: 10.1016/j.bbrc.2023.10.038.

Response: Thank you for your comment. We added a relevant paragraph in the section ‘‘The multifaceted role of apolipoprotein E in the brain’’: Previous research has specifically shown the crucial role of APOE in the metabolism of Aβ (10.1016/j.biopha.2023.116071, 10.1016/j.bbrc.2023.10.038, 10.1016/j.neulet.2023.137532,https://doi.org/10.1007/s00018-023-05026-w). Isoform dependent binding to Aβ regulates its production and clearance. APOE4 enhances Aβ production and hinders its phagocytic clearance leading to Aβ deposition. APOE2, on the other hand, decelerates this process. At the same time, the role of APOE-lipid interactions appears to be of pivotal importance in αS aggregation, as well (https://doi.org/10.1126/scitranslmed.aay3069). Neuronal APOE has been reported to attenuate both neuronal αS uptake and release, with APOE deficiency decreasing expression of APOE receptors responsible for αS uptake and enhancing chaperone-mediated autophagy (https://doi.org/10.3390/ijms23158311). APOE deficiency ultimately results in accumulation of insoluble αS and phosphorylated αS in the brain, as well as altered lipid profiles (10.1007/s00401-021-02361-9). The modification of membrane composition appears to influence αS binding which might lead to altered β-sheet formation and in turn further fibrillization (10.1007/s00401-021-02361-9).